# Does Testosterone Salvage PDE5i Non-Responders? A Scoping Review

**Mike Pignanelli [1], Noah Stern [1] and Gerald Brock [2,*]**

[1] Division of Urology, Department of Surgery, Schulich School of Medicine and Dentistry, Western University, 268 Grosvenor Street, Rm B4-657, London, ON N6A 4V2, Canada
[2] Division of Urology, Western University, London, ON N6A 4V2, Canada
* Correspondence: gebrock88@gmail.com

**Abstract:** Erectile physiology, in order to function normally, requires the complex coordination of endocrine, neurocognitive, neuromuscular and vascular mechanisms. Testosterone (T) influences male sexuality as well as penile erections at multiple levels, including a direct influence on the nitric oxide synthase (NOS)/cGMP/phosphodiesterase 5 pathway in the penis. However, the precise role of testosterone replacement (TRT) to "salvage" men with mixed ED failing phosphdiesterase-5 inhibitors (PDE5i) remains unclear. We conducted a scoping review identifying the rationale for TRT in ED failing PDE5i, and we critically discuss clinical trials that have examined TRT in the setting of PDE5i use. Overall, TRT replacement appears to be well tolerated and may enhance the response to PDE5i and quality of life, particularly for men with mixed ED, and particularly among men with very low levels of testosterone. However, most of the available literature examines concurrent TRT alone or simultaneous TRT + PDE5i usage, without necessarily selecting for PDE5i failure cases. The present studies are limited to heterogenous studies with small sample sizes, without an exact predominant etiologic factor causing ED. Furthermore, studies showing the most benefit are non-placebo-controlled trials; however, the correction of more profound hypogonadism may lead to an improved response to PDE5i. Stronger conclusions would require properly selected patient populations and larger placebo-controlled RCTs.

**Keywords:** testosterone; hypogonadism; testosterone replacement therapy; erectile dysfunction; phosphodiesterase inhibitors





## 1. Introduction

Erectile dysfunction (ED) is the inability to attain or maintain an erection adequate for sexual satisfaction [1]. Due to the sensitive nature of ED, the annual incidence is unknown and likely underreported. The prevalence of ED in America is approximately 52% of men aged 40–70 years [2], and 70% of men aged 70 years or older [3]. By 2025, it is estimated that the number of men with ED will be 322 million [4]. The requisite pressure for penile erections is achieved when there is increased arterial blood flow into the penis coupled with an increase in resistance to venous outflow. Although a simple concept, the physiology of penile erection is complex and requires proper development of the penile anatomy, a neurological response to sexual stimuli, intact neurovascular structures, and paracrine signaling cascades [5]. An understanding of this intricate interplay between neurologic, physiologic, and anatomic pathways is essential to understand the pathophysiology of erectile dysfunction and the potential role for testosterone in its management.

The dorsally paired corpora cavernosa are the functional erectile tissues of the penis. The cavernous artery, a terminal branch of the internal pudendal artery, derived from the internal iliac artery, also courses through the root of the corpora to supply the penis with arterial blood lined by smooth muscle cells. With the filling of the tunica, the emissary veins are closed by the elastic layers of the tunica albuginea, resulting in increased corporal

pressures [6]. The parenchyma of the corpora cavernosa is composed of endothelial-lined sinusoidal structures with trabecula of smooth muscle and collagen/connective tissue with a network of autonomic nerve terminals and cavernosal and helicine arteries running throughout the structure [7]. The stimuli for erection are CNS reflexes, either relaying through pudendal nerve sensory stimuli to S2-4 and activation of parasympathetic nerve fibers of the pelvic nerve, or sexually arousing imagery/thoughts from the cortex that suppress sympathetic outflow from T11-L2 through the hypogastric nerve to the penis [8]. The cavernous nerve of the penis supplies the corpora and hold both vasoconstricting sympathetic fibers and vasodilating parasympathetic fibers [6]. Finally, somatic nerve innervation by the pudendal nerve to bulbocavernosus and ischiocavernosus muscles leads to contraction and pressurizes the corpora cavernosa to further increase rigidity [9].

Chiefly studied among the many putative molecular mediators of erection is nitric oxide (NO). Ultimately, sexual stimuli likely trigger both nitric oxide release from pre-synaptic parasympathetic neuron nitric oxide synthase (nNOS) and the subsequent stretching that causes the acetyl-choline-mediated excitation of the post-synaptic nerve terminals of endothelial nitric oxide synthase (eNOS) and a feed-forward loop of vasodilation and shear forces that is mechano-transduced to produce more endothelial NO [10–12]. NO is produced from L-arginine by constitutive nitric oxide synthase (NOS) and diffuses outside of the endothelium to reach nearby smooth muscle cells to work as a functional syncytium, which can also produce autocrine NO via inducible NOS. Once NO diffuses into the arteriolar smooth muscle cell cytoplasm, it activates cytosolic guanylate cyclase to increase the concentration of intracellular cyclic guanosine monophosphate (cGMP). Intracellular pathways such as the inositol-triphosphate (IP3) and diacyl-glycerol (DAG) pathways lead to a reduction in intracellular calcium levels that promotes cellular relaxation. Furthermore, protein kinases are activated to phosphorylate myosin light chains, which leads directly to smooth muscle relaxation and an increase in arterial flow to the corpora [12]. The erection then persists until stimulation ceases, and the sympathetic nervous system is involved in promoting the detumescence of the penis [13]. Though likely an over-simplification, detumescence requires the metabolism of cGMP to GMP, achieved by phosphodiesterase (PDE) isoenzymes, of which PDE type 5 is the predominant type identified in the human corpora cavernosa [11]. A reduction in PDE5 activity and subsequent increase in cGMP accumulation in the relaxed penile arterial smooth muscle provides the physiologic basis for the efficacy of competitive PDE5is, i.e., sildenafil, tadalafil, avanafil and vardenafil [14,15]. Oral PDE5is have been extensively studied and are the starting point in the pharmacological treatment of all etiologies of ED [16–19].

Given their favorable safety profile and ease of prescription, PDE5is are commonly prescribed for ED. For example, it is estimated that over 35 million patients filled a prescription for sildenafil alone between its release in 1998 and September 2007, when the post-marketing safety database was closed [16]. A recent meta-analysis of 103 studies and 26,845 patients suggests that the efficacy of PDE5i may be as high as 80% in placebo-controlled trials [20]. However, the discontinuation rate of PDE5i remains high at 4% per month (50% after one year). PDE5i discontinuation is most often due to a lack of efficacy or partner-related problems and is correlated with comorbid conditions such as hypertension, obesity, and hypogonadism [21].

Testosterone is an anabolic steroid hormone produced by Leydig cells of the testicle and cells of the zona reticularis in the adrenal gland. Serum testosterone is carried primarily by sex-hormone-binding globulin, and bioavailable testosterone (free and loosely bound by albumin) is converted to metabolite dihydrotestosterone via 5 alpha-reductase in peripheral tissues, including cells of the corpora cavernosa [22,23]. Endogenous production of testosterone is under the control of the pulsatile diurnal release of gonadotropin-releasing hormone (GnRH) produced by the hypothalamus, subsequently leading to luteinizing hormone (LH) release from the anterior pituitary [24]. There is evidence that testosterone is involved in maintaining male sexual characteristics, muscle mass, erythropoiesis, and fertility, and is important throughout the sexual cycle of libido, erection, orgasm, ejaculation,

and detumescence [25]. The seminal *T Trials* demonstrated an increase in libido and sexual interest in men with T < 9.4 nmol/L (<270 ng/dL) and this finding was proportional to an increase in levels throughout the study. Overall, greater effects on sexual function rather than erection were seen, but the greatest improvement in ED was shown if patients were severely hypogonadal (<8 nmol/L) or diabetic/obese [26].

Partial androgen decline in the aging male (PADAM or ADAM) is thought to be due to reduced gonadal function leading to decreased serum testosterone levels [27]. Despite initial concerns surrounding the risk of CVD/VTE with testosterone replacement, the current literature suggests that hypogonadism is associated with increased risks of heart attack, stroke, and complications of accelerated atherosclerosis, and there is a paucity of high-quality data to suggest that testosterone replacement in hypogonadal men leads to an increase in clinically significant prostate cancer [28]. The clinical syndrome of hypogonadism is defined by the Endocrine Society as men with signs and symptoms of testosterone deficiency and unequivocally and consistently low morning total serum testosterone less than 364 ng/dL (12 nmol/L) and/or free testosterone concentrations when indicated of ≤220 pmol/L, depending on the assay or laboratory used [29]. However, it should be noted that the cut-off value for hypogonadism is controversial. Other societies, such as the America Urology Association, endorse a cut-off of total testosterone of less than 300 ng/dL or 10.4 nmol/L [28], and the International Society of Sexual Medicine suggests a value consistently <231 ng/dL or <8 nmol/L [29]. The Endocrine Society (ES), American Urology Association, Canadian Urology Association, and European Urology Association have all published guidelines for treating patients with hypogonadism [28–31]. In all of the cited guidelines, the authors recommend two AM testosterone levels, ideally measured via liquid chromatography–mass spectrometry as it is the most sensitive, specific, and precise method for the quantification of testosterone levels [22]. Further, the guidelines recommend against routinely screening for low testosterone in asymptomatic men or men with acute illness and ordering LH/FSH to rule out secondary hypogonadism, PSA (>40 years), and hematocrit.

Once primary hypogonadism is diagnosed via high LH and low T levels in a symptomatic individual, TRT can be considered. Contraindications to TRT include a high risk for or active prostate cancer, unevaluated positive DRE, PSA > 4 ng/mL, unevaluated polycythemia, 3–6 months from an acute cardiovascular event, and/or a desire for fertility. Testosterone can be given via the oral, transdermal (patch, gel, buccal), subcutaneous, intramuscular, or intranasal routes. Alternatives to directly providing exogenous testosterone include the use of selective estrogen receptor modulators, hCG, or aromatase inhibitors to increase the endogenous levels of testosterone. Target testosterone should approximate the normal physiologic range of 450–600 ng/dL according to the AUA, with repeat lab testing 3–6 months after starting TRT. Depending on the modality of TRT, repeat testing may be as short as 2–4 weeks (topical/nasal) to a mid-way 10-week IM injection cycle (see Table 7 in the unabridged AUA guidelines for more details). Should the patient not reach the target threshold, a change in modality can be considered. The timeline of therapeutic benefits includes 3 months for libido, energy, and mood/neurocognitive effects (assuming optimized from confounding conditions) and 6 months for an improvement in erections, and improvements in anemia, bone density, and lean muscle mass. Should signs and symptoms remain unaddressed by TRT, it can be discontinued. Testing for PSA should continue as in any eugonadal man, with monitoring for polycythemia every 6–12 months [28,29,31].

Hypogonadism and ED are frequently comorbid conditions and it is generally accepted that androgens modulate the erectile physiology [32]. Historically, normal erectile function is thought to require a "threshold testosterone level" of around 10–12% of the normal lower limit physiological level (approximately 30 ng/dL), with a dose-dependent worsening of ED below this threshold and an improvement in erectile function with androgen replacement [32,33]. Hypogonadism with a cut-off of <300 ng/dL has been suggested as a risk factor for PDE5i failure [34], but ED is often comorbid with current smoking status, advanced age, metabolic syndrome, obesity, sedentary behavior, and type 2 diabetes. Furthermore, other studies have shown that PDE5is still are able to improve erection scores,

regardless of baseline hypogonadal status being close to physiologically normal scores [35]. This review will focus on clinical trials that have been conducted in humans investigating whether testosterone replacement may fit into the ED paradigm after a trial PDE5i prior to declaring treatment failure. We will also offer expert opinions on whether amendments should be made considering the available evidence.

## 2. Materials and Methods

A scoping review was conducted using Medline, PubMed, EMBASE, and Cochrane searches, including the following searches between 31 December 1992 and 22 October 2022: ("Testosterone" OR "Hypogonadism" OR "Androgen deficiency" OR "Castrate*"OR "Androgen deprivation") AND ("Erectile dysfunction" OR "Sexual dysfunction" OR "PDE5" OR "PDE5i" OR "Phosphodiesterase" OR "Viagra" OR "Sildenafil" OR "Tadalafil" OR "Cialis"). For relevant review articles identified during manual title and abstract screening, we proceeded to examine their full-text contents. Relevant primary research and clinical trials examining both hypogonadism and PDE5i use for ED were identified and prior reviews on this topic were gathered from their references, and we ensured that they were included in the full-text review. Covidence (www.covidence.org), a web-based collaboration software platform that streamlines the production of systematic and other literature reviews, was used to facilitate the iterative development of this review. Two reviewers independently screened and reviewed all articles according to prespecified criteria. Conflicts were reviewed and disagreements were mutually resolved post-discussion.

## 3. Results

The search strategy identified 160 studies for screening. After removing duplicates, 74 studies were identified for title and abstract screening. Of these, 23 were deemed to be irrelevant to the research question after screening titles and abstracts. Three studies were excluded due to unavailability of the full-text PDF file. After a full-text review, 41 studies were excluded. Ultimately, nine studies met all inclusion criteria, investigating the role of testosterone in salvaging PDE5i non-responders. Four of the studies identified were prospective randomized control trials, three were cohort studies, one was a crossover study, and one utilized a sequential design (Figure 1).

These articles included a total of 675 male subjects with a mean age of 54.9. Testosterone formulations were transdermal in four studies, oral in three studies, and intramuscular in one study. Six studies used 25–100 mg sildenafil on demand, two used 5–10 mg tadalafil, and one deferred to the PDE5i preferred by the patient. All studies utilized the International Index of Erectile Function (IIEF), with a specific focus on the erectile function domains (Table 1).

The etiology of erectile dysfunction was described in two of the eight included studies. Seven of the eight studies monitored and confirmed testosterone improvements with therapy. All showed improvements to specified normalized values. One of the eight studies reported no improvement in PDE5i response with testosterone replacement (Table 1).

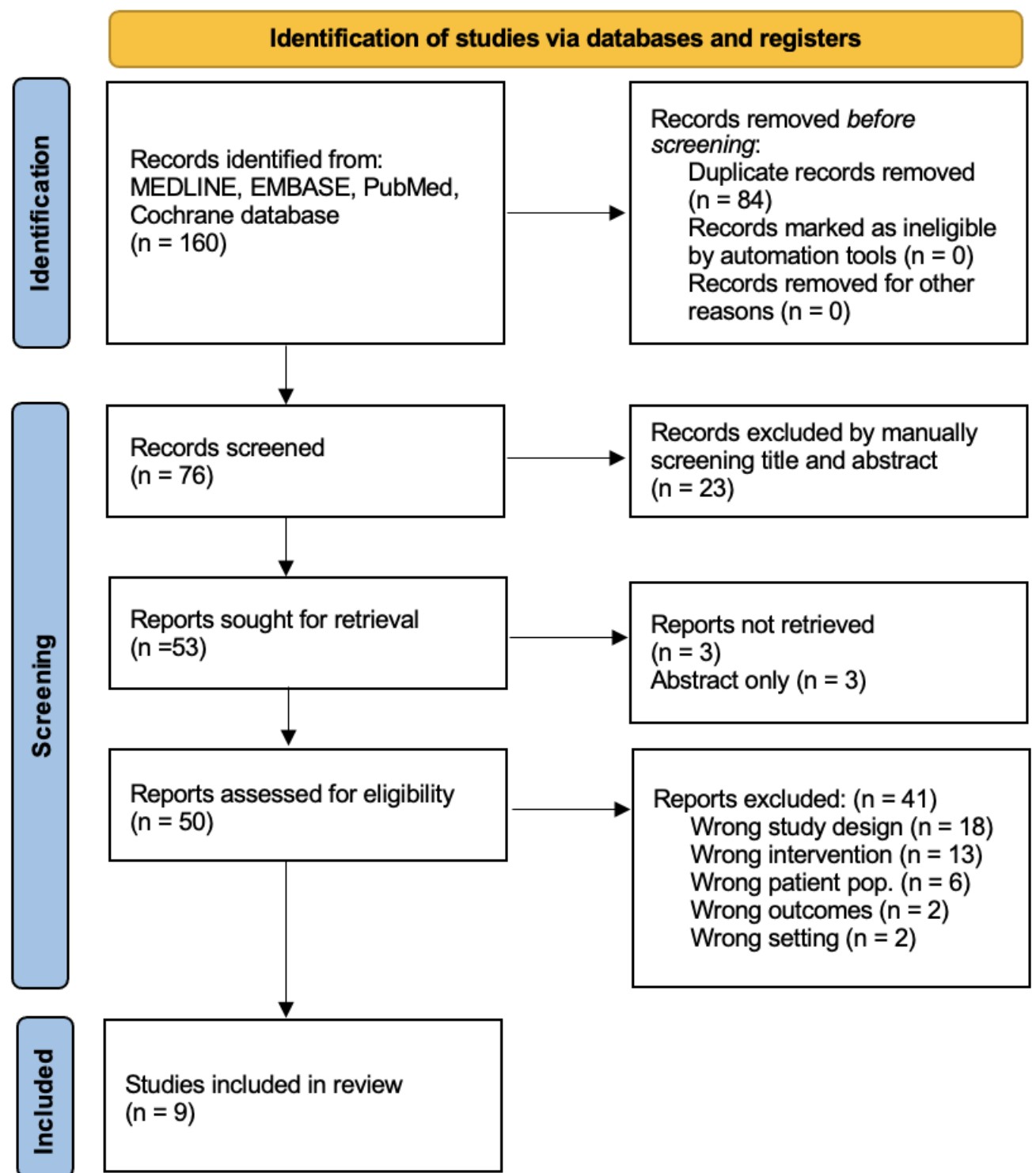

**Figure 1.** PRISMA table outlining results of literature search.

**Table 1.** Summary of included studies in hypogonadal men with inadequate response to PDE5i.

| Study, Year | n | Study Type, Country | Testosterone Formulation | PDE5i | Control | Duration of Treatment (Weeks) | Outcome Measure |
|---|---|---|---|---|---|---|---|
| * Kalinchenko 2003 [36] | 120 | Cohort, Russia | Oral testosterone undecanoate (120 mg/day) | Sildenafil (100 mg PRN) | | 8 | IIEF |
| * Aversa 2003 [37] | 20 | RCT, Italy | Transdermal patch (5 mg/day) | Sildenafil (100 mg PRN) | Placebo | 10 | IIEF |
| Shabsigh 2004 [38] | 75 | RCT, USA | 1% testosterone gel | Sildenafil (100 mg PRN) | Placebo | 14 | IIEF |
| Shamloul 2005 [39] | 40 | Prospective cohort, Egypt | Oral undecanoate (120 mg/day) | Sildenafil (50–100 mg PRN) | | 12 | PADAM, IIEF |
| Hwang 2006 [40] | 32 | Cohort, Taiwan | Oral undecanoate (160–240 mg/day) | Sildenafil (100 mg PRN) | | 4 | IIEF |
| Buvat 2011 [41] | 173 | Multi-centered RCT, European | 1% hydroalcoholic T gel (up to 10 mg) | Tadalafil (10 mg/day) | Placebo | 8 | IIEF |
| Garcia 2011 [42] | 29 | Cohort, Venezuela | Parenteral testosterone undecanoate (2 injections q6 weeks, followed by 12-weekly injections) | Unspecified | | 102 | IIEF |
| † Spitzer 2012 [43] | 140 | RCT, USA | 1% transdermal testosterone gel (10 g/day) | Sildenafil (25–100 mg PRN) | Placebo | 12 | IIEF |
| Kim 2013 [44] | 46 | Cohort, Korea | Parental testosterone enanthate q4 weeks | Tadalafil (5 mg/day) | | 36 | IIEF |

* Etiology of ED was presumed to be arteriogenic in Aversa study and complication of T2DM in Kalinchenko study. † No further improvements seen in PDE5i + testosterone compared to PDE5i + placebo.

## 4. Discussion

Of the nine studies included in this full-text review, only two studies were able to categorize the etiology of erectile dysfunction and only one study reported no improvement with combination therapy compared to PDE5i alone. Spitzer et al.'s 2012 RCT randomized patients to 7 weeks of sildenafil followed by 14 weeks of testosterone gel or placebo. No change in IIEF (12.2 vs. 12) was observed between groups; however, the initial PDE5i trial improved scores by at least 7.7 for all participants [43]. However, these patients were not necessarily PDE5i failures as they showed increased IIEF scores from baseline; however, patients were still categorized as having moderate ED, with a score of 12. The included patients had a high degree of comorbidities that were not optimized during the study, including hypertension, diabetes, and obesity. These comorbidities did appear to be well balanced between the study groups. The authors of this study therefore concluded that patients who have a moderate degree of initial success with PDE5i use may not expect a further benefit from testosterone replacement due to the presence of other etiological ED factors. A 2005 cohort study by Shamloul et al. similarly showed that combination therapy improved IIEF scores in PDE5i non-responders (IIEF 10.1 to 15) as compared to PDE5i partial responders (IIEF 15.3 to 17.5) [39]. Shabsigh et al. performed a 2004 RCT that showed that patients on combination therapy had improved erectile function at week 4 (4.4 vs. 2.1); however, the statistical significance of this trend was lost by the end of the study [38]. Similarly, no difference was seen between groups regarding the proportion of IIEF responders (response of 4 or 5 to questions 3 and 4 of IIEF) or improvers (response increased by at least one category for either question 3 or 4 of IIEF as compared to baseline).

Buvat et al.'s 2011 RCT found that a monitored and optimized PDE5i trial was able to salvage up to 56% of patients initially deemed PDE5i non-responders [41]. In all patients, this RCT found no difference between the experimental combination arm and the control; however, stratifying by baseline testosterone values showed marked improvements in patients with testosterone less than 3 ng/mL (increased EFD 6.18 vs. 2.33, SEP3 33.1 vs. 13.4). Kim et al. also stratified patients by hypogonadal state, showing that men with more severe testosterone deficiency (<250 ng/dL) had a superior response rate to combination therapy (57.9%) as compared to those with moderate deficiency (250–350 ng/dL), where only 16% responded [44]. Both the moderate and severe groups showed improvements in erectile function domain scores, increasing from 8.3 to 12.4 and from 7.2 to 16.5, respectively. Interestingly, the severe group maintained improved erectile function scores (13 vs. 7) during the washout period, where testosterone levels returned to baseline. A comparable randomized control trial by Aversa et al. compared testosterone gel vs. placebo in PDE5i unresponsive men with arterial insufficiency as a cause for erectile dysfunction [37]. The erectile function domains of the IIEF improved from 14.4 to 21.8 in the treatment arm as compared to 13.2 to 14.2 in the placebo group. Statistically significant improvements were also seen in intercourse satisfaction and overall satisfaction domains.

A 2003 cohort study by Kalinchenko et al. showed that transdermal testosterone was able to salvage 70% of patients who were PDE5i non-responsive, increasing sexual contact from 0.5 to 3–4 times per month [36]. After a washout period, non-responsive rates returned to baseline. Garcia et al. showed that intramuscular testosterone improved sexual desire (4.1 to 7.2) and erectile function (9.1 to 26.2) by 30 weeks [42]. Interestingly, 17% of their patients felt that their erections had improved to the point where PDE5is were no longer needed.

Finally, Hwang et al. identified 32 hypogonadal men with erectile dysfunction and trialed 30 days of PDE5i followed by 30 days of testosterone, followed by combinations of PDE5i and testosterone therapy [40]. This group found that 34% of patients who failed to respond to PDE5is achieved satisfactory improvements with testosterone alone, while a further 37.5% responded to combination therapy. Overall IIEF scores improved from 12.6 at baseline to 12.0 on sildenafil alone, to 14.8 on testosterone alone, and to 17.5 on combination therapy. Ratings of participants' ability to achieve and maintain erections also improved from 2.4 to 3.5 and from 2.3 to 3.6 on combination therapy.

Possible physiologic explanations for the underlying mechanism behind these clinical findings can be traced back to prior animal models. Though testosterone may also influence the neural mechanisms of erection and prevent the apoptosis of penile cavernosal smooth muscle cells, testosterone is directly involved in the expression and activity of NOS in the corpora cavernosa [45–48]. Castrated animals with low testosterone also show decreased NOS as well as PDE expression and activity in vivo. Furthermore, a minimal erectile response is seen with PDE5i when cavernous nerve stimulation is applied without androgen repletion in apomorphine rodent models [49–51]. It is thought that testosterone may modulate both NOS and PDE5 activity, as increasing NO synthesis requires a counterbalance to maintain homeostasis [48]. In partially viable cavernous nerve injury models, chronic PDE5i also seems to improve NOS expression in the endothelium of the rat penis, possibly through the VEGF-Akt pathway or by decreasing reactive oxygen species [52,53]. However, the androgen receptor itself stimulates NOS in a possible alternative pathway in endothelial cells though activation of the c-Src/IP3/Akt pathway. In summary, animal models of testosterone replacement may enhance the effect of PDE5i by increasing endothelial NOS production and possibly nervous tissue NOS [52]. With greater NOS expression produced by both PDE5i and testosterone, theoretically, greater amounts of NO can be generated, leading to an increase in cGMP production in target tissues.

Strengths of this review include providing an overview of the existing evidence in the PDE5i failure field across multiple databases (Pubmed, EMBASE, and Cochrane database). We had two reviewers independently screen the results of an inclusive search strategy and inspect the full-text articles collected. The Covidence software was used to facilitate the iterative development of our review. As with any scoping review, limitations include no risk of bias assessment, given that this was a scoping review. Given the findings of the reviewed literature, more dedicated, high-quality primary studies examining testosterone replacement for PDE5i failure patients are required prior to a formal systematic review being conducted. Another limitation to this review is that not all studies described the etiology for the patients' ED, and this may influence the follow-up time required to see the effect of TRT. For example, some authors suggest that for TRT to salvage ED from a venous leak, it may take 6 months [54] or even >12 months to see an improvement [55]. International guidelines also support the role of TRT for an improvement in erectile function after a trial of PDE5i in men who had symptoms and hypogonadal testosterone levels [1,30,56]. The benefits of salvaging a patient with TRT compared to moving on to injection therapy include several follow-up opportunities to address the systemic health of the patient, patient education (diet, exercise, smoking cessation, and lifestyle), addressing other areas of the sexual cycle, avoiding the discomfort or complications (corporeal fibrosis/peyronies, hematoma, priapism) of intracavernosal injection therapy, and a prolonged time to ED surgery.

## 5. Conclusions

There is reasonable evidence in the literature suggesting that a combination of PDE5i and testosterone may be superior to PDE5i alone; however, this is the first review to investigate the evidence for the use of testosterone in salvaging PDE5i non-responders. From the available evidence, profoundly hypogonadal PDE5i non-responders appear to obtain the most benefit from combination therapy, with the potential to salvage up to 70% of PDE5i non-responders. However, it remains unclear which etiologies of ED will respond better to PDE5i after TRT. Studies identified in this review are limited by small samples, relatively short follow-up times, and multifactorial ED patient populations with many comorbidities and TRT dosing strategies. For asymptomatic hypogonadal patients with multifactorial ED and comorbidities such as hypertension, diabetes, and obesity, we suggest that expectations in erectile function gained solely from the pharmaceutical treatment of hypogonadism should be tempered. We recommend undergoing shared decision making prior to measuring testosterone and subsequent supervised TRT in patients with ED failing PDE5is due to the possible recovery of PDE5i sensitivity in some patients, as identified by studies in this review.

**Author Contributions:** All authors contributed to all aspects. All authors have read and agreed to the published version of the manuscript.

**Funding:** This research received no external funding.

**Institutional Review Board Statement:** Not applicable.

**Informed Consent Statement:** Not applicable.

**Data Availability Statement:** Not applicable.

**Conflicts of Interest:** The authors declare no conflict of interest.

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
