# Peer review of "Does Testosterone Salvage PDE5i Non-Responders? A Scoping Review"

_endocrines, doi:10.3390/endocrines4010011_

Round 1
Reviewer 1 Report
It is my pleasure to review this paper entitled “Does Testosterone Salvage PDE5i non-responders?”
This is a scoping review identifying the rationale for TRT in ED failing PDE5i.
Overall article is well written English is fluently and adequate. However, there are some drawbacks that could be addressed before an eventual publication.
Please add in the title the nature of the study.
Line 30 please add a reference
Please add information about incidence and prevalence of erectile dysfunction, use of pde5i and epidemiology of pde5i non-responder patients
Please separate results from discussion
Please add strengths and limitations of the paper
Authors could improve the paper with a recent publication doi: 10.3390/ijms23073535.
Author Response
Comments and Suggestions for Authors
It is my pleasure to review this paper entitled “Does Testosterone Salvage PDE5i non-responders?”
This is a scoping review identifying the rationale for TRT in ED failing PDE5i.
Overall article is well written English is fluently and adequate.
- Thank you
However, there are some drawbacks that could be addressed before an eventual publication.
Please add in the title the nature of the study.
- Changed title to “Does Testosterone Salvage PDE5i non-responders: a Scoping Review.”
Line 30 please add a reference
- See lines 28-32. Added epidemiology of ED (as requested below) and added references.
Please add information about incidence and prevalence of erectile dysfunction, use of pde5i and epidemiology of pde5i non-responder patients
- Incidence and prevalence of ED?
- Incidence is difficult to accurately determine due to sensitive nature of condition and is therefore likely under-reported.
- Prevalence of ED
- Fung, M.M., Bettencourt, R., Barret-Connor, E., & Sullivan, M/D. (2017). Prevelance and incidence of erectile dysfunction in the United States. The Journal of Sexual Medicine, 14(5), 643-650.
- Liu, L., Dong, Y., & Zhang, L. (2018). The epidemiology of erectile dysfunction: Prevalence and risk factors. Current Sexual Health Reports, 10(2), 97-106.
- Use of PDE5i?
- Included historical prescription rates during post-marketing surveillance of sildenafil (Giuliano et al 2010) as an example for how common PDE5i prescription may be, although prescription patterns of PDE5i are not a focus of this present study. Also, amongst clinicians it is well known that PDE5is are very commonly prescribed for ED. (Lines 86-89)
- Epidemiology PDE5i failure?
- Included efficacy of PDE5is as per recent meta-analysis – ~80% (Madeira et al 2021) as well as discontinuation rate - ~4% per month (Corona et al 2016) (lines 89-94)
Please separate results from discussion
- Results: lines 189-205.
- Discussion: lines 208-325
Please add strengths and limitations of the paper
- Strengths line 332-336
- Limitations and future direction: 336-351
Authors could improve the paper with a recent publication doi: 10.3390/ijms23073535.
- Thank you for suggestion. Added reference to introduction (line 114-115).
Reviewer 2 Report
The manuscript entitled "Does Testosterone Salvage PDE5i non-responders?" is interesting and well written generally.
1. The conclusion with how this review would be helpful to clinicians should be mentioned. Please suggest the positioning of testosterone measurement for the non-PDE5i responders in daily clinical practice. In addition, is testosterone therapy recommended for all non-PDE5i responders? Or limited for the hypogonadal cases?
2. In table 1, information regarding country, target population (hypogonadal men or not), and brief results should be included, which is more helpful for readers to understand results of this review.
3. In line 172 (and in figure 1), 8 studies met all inclusion criteria; however 9 studies are shown in table 1.
4. In line 184, ‘Spitzer et al’s 2005’; Is this "Spitzer et al’s 2012"?’
5. In line 189, ‘PED5i partial responders’; this is misspelling.
6. Only one study reported no improvement with combination therapy. Please provide details about this report. Were subjects included in this study hypogonadal men? What was the cause of ED in the target subjects?
Author Response
The manuscript entitled "Does Testosterone Salvage PDE5i non-responders?" is interesting and well written generally.
- The conclusion with how this review would be helpful to clinicians should be mentioned. Please suggest the positioning of testosterone measurement for the non-PDE5i responders in daily clinical practice. In addition, is testosterone therapy recommended for all non-PDE5i responders? Or limited for the hypogonadal cases?
- Positioning of testosterone measurement/TRT recommendation for all non-PDE5i responders?
- See lines 401-407.
- Positioning of testosterone measurement/TRT recommendation for all non-PDE5i responders?
- In table 1, information regarding country, target population (hypogonadal men or not), and brief results should be included, which is more helpful for readers to understand results of this review.
- Added country of all centers
- Target population was all hypogonadal. This was part of exclusion criteria (i.e. men who were not found to have PDE5i and did not have low testosterone were not offered supra-therapeutic testosterone).
- See updated table – Lines 384-387.
- In line 172 (and in figure 1), 8 studies met all inclusion criteria; however 9 studies are shown in table 1.
- Error in automated removal of duplicated records manually corrected (84 not 86).
- Manual input error upon transcribing “records excluded by title and abstract” into Prisma flow diagram (23, not 32).
- Consolidated 3 reports that returned as abstract only (2 from original duplicate record removal + 1 identified previously as abstract only)
- We reviewed reports assessed for eligibility and reason for exclusion. On the software used, 33 papers were marked as reviewed by both authors, however 9 others were marked as excluded by author 1 and not author 2 on covidence.org and did not get included in the reports excluded count. 1 article was re-categorized in “abstract only” as described above.
- For transparency, a full list of included and excluded references is available, upon request.
- New figure inserted on line 354-381.
- In line 184, ‘Spitzer etal’s 2005’; Is this "Spitzer et al’s 2012"?’
- Yes, double checked reference and manual error (Shabsigh et al was the 2005 paper, correct paper for sentence is Spitzer 2012).
- In line 189, ‘PED5i partial responders’; this is misspelling.
Error remedied.
- Only one study reported no improvement with combination therapy. Please provide details about this report. Were subjects included in this study hypogonadal men? What was the cause of ED in the target subjects?
- See lines 211 – 222. All men who were not hypogonadal were excluded. Added further comments on 215-222 regarding testosterone replacement not further improving gains of PDE5i as it does not necessarily (in isolation) improve other comorbid conditions.
Round 2
Reviewer 1 Report
Authors improved paper according to suggestions